# Perceived Vaccine Availability and the Uptake of Measles Vaccine in Sudan: The Mediating Role of Vaccination Hesitancy

**DOI:** 10.3390/vaccines10101674

**Published:** 2022-10-08

**Authors:** Majdi M. Sabahelzain, Ahmed Tagelsir, Yasir Ahmed Mohammed Elhadi, Omayma Abdalla

**Affiliations:** 1Public Health Training and Research Unit, Nutrition and Health Center for Training and Research, School of Health Sciences, Ahfad University for Women, Omdurman P.O. Box 167, Sudan; 2Clinical Pharmacy Division, Aldoha Specialized Hospital, Khartoum P.O. Box 303, Sudan; 3Public Health Department, Sudanese Medical Research Association, Khartoum P.O. Box 303, Sudan; 4Expanded Program on Immunization, Federal Ministry of Health, Khartoum P.O. Box 303, Sudan

**Keywords:** vaccine access issues, vaccine uptake, vaccine hesitancy, Sudan, measles, mediation analysis

## Abstract

This study aimed to evaluate whether measles vaccine uptake can be predicted directly or indirectly by parental perceptions about the availability of measles vaccine services with parental hesitancy towards the measles vaccine as a potential mediator. This was a community-based cross-sectional study conducted at Omdurman locality in Khartoum state, Sudan in February 2019. The study population included parents/guardians having at least one child aged 2–3 years old. Mediation analysis was conducted using two models, the ordinary least squares path analysis and multiple logistic regression. These models considered perceived vaccine accessibility and availability as independent factors, vaccine uptake as dependent factors, and vaccine hesitancy (PACV scores) as a mediator. A total of 495 responded and the mean age of the mothers who participated in the study was 31.1 (SD = 5.73). Half of the respondents (50.1%) completed university education and nearly three-quarters of the respondents (74.7%) were housewives. After controlling for the other factors, including the mother’s age and the number of children, parental perception about the accessibility and availability of the measles vaccine influences the uptake of the measles vaccine indirectly through the mediation effect of measles vaccine hesitancy. We suggest that intervening in measles vaccine hesitancy in addition to measles vaccination access issues will have a positive impact on the uptake and coverage of the measles vaccine in Sudan.

## 1. Introduction

Globally, it is estimated that measles vaccination has averted about 23.2 million deaths between 2000 and 2018 [1]. Vaccine uptake is one of the indicators that is used to inform and guide the immunization program worldwide [1,2,3]. Previous research has shown that measles vaccination coverage is affected by many factors pertaining to service quality, delivery, and access barriers [4]. However, studies have not consistently identified the direct or indirect impacts of the perceived or real access issues on the uptake of the vaccine [5,6].

Findings from studies conducted in low- and middle-income countries (LMICs) including Sudan have shown that factors related to facility-level and service providers have an impact on vaccination coverage. This includes the reluctance of healthcare providers to open a 10-dose unpreserved vial in situations if immunization sessions have fewer than 6–8 infants in the clinic, in order to comply with the multidose vial policy of discarding the ten-dose vial after six hours from opening the vial [7,8,9,10,11,12]. Although this practice can reduce vaccine wastage, it is associated with missed opportunities for vaccination and vaccine hesitancy, as many parents are actively trying to get their children vaccinated against measles but are turned away when the vaccinators refuse to open the measles vaccine vial [7,13,14].

Vaccine hesitancy is considered one of the most important predictors of vaccine uptake. It has been listed by the WHO among the top ten threats to global health [15]. The impacts of vaccine hesitancy on vaccine uptake and demand are poorly understood in LMICs, suggesting a more complex relationship between supply-side and demand-side factors than in high-income countries [6,12,16].

The WHO Strategic Advisory Group Experts on immunization (SAGE) described vaccine hesitancy as a delay in acceptance or refusal of vaccination, despite the existence of vaccination services. This behavior is influenced by some factors, such as complacency (perceived risks of vaccine-preventable diseases are low and no vaccines are needed), convenience (access issues and constraints), and confidence [17].

Measles is considered the third leading cause of mortality among children under five and the first among vaccine-preventable diseases in Sudan [18,19]. The national measles vaccination coverage is the lowest among all child vaccinations (i.e., for the first and the second dose of measles-containing vaccine, 88% and 72%, respectively) and is lower than the required level of 95% coverage for measles elimination [20]. Previous studies in Sudan showed the existence of measles vaccine hesitancy is attributed to different drivers including vaccination access-related issues, such as the parental perception that measles vaccination services are unavailable and inaccessible [7,21]. Additionally, a study found that the uptake of the measles vaccine among children is predicted by their parental hesitancy toward the measles vaccine [22].

We assume in this study that perceived access issues including unavailability of vaccination services may play an important role in predicting uptake of measles vaccine either directly or through measles vaccination hesitancy (Figure 1). The accessibility and availability of the measles vaccine (perceived or real) were widely debated as an access issue (barrier) rather than psychological status [4]. To our knowledge, there are limited data in Sudan about the predictors of measles vaccine uptake. Therefore, in this study, we aimed to evaluate whether measles vaccine uptake can be predicted directly or indirectly by parental perceptions about the availability of measles vaccine services with parental hesitancy towards the measles vaccine as a potential mediator.

## 2. Materials and Methods

### 2.1. Study Design

The research design was a community-based cross-sectional study and was conducted in two urban districts in Omdurman locality in Khartoum state in February 2019. These two districts were selected for the study because they reflect the typical sociodemographic and socio-cultural situation in Sudan. As the two are in an urban setting, this may ensure exposure to vaccination communication campaigns as well as the availability of vaccination services. The latter is a prerequisite for the assessment of vaccine hesitancy.

### 2.2. Population and Sampling

#### 2.2.1. Population

The study population included parents/guardians having at least one child aged 2–3 years old. Either mothers or fathers were eligible for participation. If there was more than one child in the same age range in the family, the parents/guardians were asked to answer about only the youngest one to reduce the risk of recall bias. If both mother and father were available, they were asked to nominate one of themselves to complete the questionnaire.

#### 2.2.2. Sampling

This study is part of a large research project about measles vaccine hesitancy in Sudan [21,22]. The sample size was calculated for the whole research using a power analysis for the association between measles vaccine hesitancy and the measles vaccination status (outcome), which showed that at least 386 participants were needed to yield an 80% power to detect an odds ratio of 1.7 at alpha level (5%). We assumed the prevalence of the outcome, the measles vaccination status among the exposed group (hesitant parents), was 50% [23]. To cover for possible drop-out due to missing information on the important questions during the survey, we recruited more participants to complete a total of 500 participants (parents/caregivers) in the study.

To ensure that people from various socio-cultural and socioeconomic backgrounds (i.e., education and wealth level) were included in this study as well as ensuring the relative availability of vaccination services as a prerequisite for assessment of vaccine hesitancy, we collected data from parents/caregivers in two different urban districts in Omdurman, Alsharafia (Wad Nubawi’s administrative unit) and Abo Saeed (Abo Saaeed’s administrative unit).

Parents/caregivers were selected in each district using consecutive sampling (convenience sample), as every parent/caregiver meeting the criteria of inclusion (had a child in the age range) was included in the study until the required sample size was achieved from each district.

### 2.3. Data Collection

Data were collected using a pre-tested, structured questionnaire with close-ended questions. Data were collected by eight well-trained graduate female students from Ahfad University for Women. The questionnaire was in the Arabic language and self-administered, but when needed, the data collectors helped the participants to fill out the questionnaire.

### 2.4. Measurements

#### 2.4.1. Dependent Variable

The dependent variable in this study was the measles vaccine uptake by the youngest child in the age range of 2–3 years (i.e., the measles vaccination status), which was measured as either fully vaccinated with two doses, where the first one is scheduled at 9 months from birth and the second between 18–24 months, or partially/unvaccinated, i.e., single dose or no dose. First, we asked the parents/guardians to show the vaccination card of their youngest child (2–3 years). If there was no card, then we asked them to report their child’s measles vaccination status.

#### 2.4.2. Independent Variables

In this study, parental perception about accessibility and availability of measles vaccination services was the main independent variable and was measured using a five-point Likert scale ranging from strongly agree to strongly disagree.

We hypothesized that measles vaccine uptake can be predicted directly or indirectly by parental perceptions about accessibility and availability of measles vaccination services with parental hesitancy towards the measles vaccine as a potential mediator. We used the Parents Attitude about Childhood Vaccination (PACV) to measure measles vaccine hesitancy. The PACV is a validated questionnaire used to predict and identify vaccine-hesitant parents and accordingly provide means of interventions. This includes 15 items categorized into three domains: immunization behavior (items 1 and 2), perceived safety and efficacy (items 7–10), and general attitudes and trust (items 3–6 and 11–15). Items in this scale were scored using a five-point Likert scale ranging from strongly agree to strongly disagree. PACV items were summed to a total score ranging from 0 to 30. Parents with children who answered “don’t know” in the behavior items of the scale (items 1 and 2) were considered missing data, because this response likely reflected poor vaccination recall rather than immunization hesitancy, as suggested by other studies. The total raw hesitancy score was converted to a 0–100 scale [24,25]. The Arabic version of the PACV scale was pre-tested with 16 parents at the Ahfad Family Center (Primary health care center) to assess its understandability and clarity of questions. The reliability and validation test was carried out and described by Sabahelzain et al. [26]. The Cronbach’s alpha was computed for this scale (Q3–Q15), which was 0.62 [26] (Appendix A).

Other independent variables that were considered as potential confounders include sociodemographic characteristics of the family, including the family’s income level (self-ranking), mother’s age, mother’s education and employment, and the total number of children in the family.

### 2.5. Statistical Analysis

Data analysis was performed using Statistical Package for social sciences (SPSS) (V 24, IBM, Armonk, NY, USA). Frequencies were generated for the sociodemographic characteristics of the family and parents’ perception of measles vaccination. Frequencies of the PACV items were calculated. Chi-square test and Fisher’s exact test (when the count in the cells is less than 5) were ran to identify factors univariate associated with the dependent variable (i.e., measles vaccination status). A *p*-value of less than 0.05 was considered statistically significant.

#### 2.5.1. The Mediation Analysis

The mediation analysis was performed using the PROCESS macro (v. 3.5.3) for SPSS (V. 24) model 3 (i.e., Simple Mediation) that was proposed by Hayes [27]. Hayes described the objective of doing simple mediation model as “to demonstrate how a variable’s effect on an outcome can be partitioned into direct and indirect effects that can be quantified.” [27]. The regression model was tested to investigate whether the association between the parental perceptions about accessibility to measles vaccination services and uptake of measles vaccine among children is mediated by measles vaccine hesitancy (i.e., PACV scores). The ordinary least squares regression model (i.e., the first model) was used to test whether the proposed mediator is predicted by parental perception about accessibility to measles vaccination services. In the second model, multiple logistic regression was used to identify the predictors of measles vaccination.

The significance of the indirect effects was analyzed using bootstrapping, with 10,000 samples and 95% confidence intervals (CIs). The mediating effects were deemed statistically significant if zero was not included in the 95% CI. The statistical significance was set at *p* < 0.05.

#### 2.5.2. Ethical Consideration

The study was approved by the Ahfad University for Women’s Review Board (IRB) and the National Health Research Ethics Committee at the Federal Ministry of Health in Sudan (No. 1-1-2018, on 30 January 2018). Written informed consent was obtained from each of the participants.

## 3. Results

### 3.1. Descriptive Statistics and Associations between the Sociodemographic and Parental Perceptions of the Measles Vaccine with the Uptake of the Measles Vaccine

As shown in Table 1, of the 495 respondents, 69.3% were from Abo Saeed. The mean age of the mothers who participated in the study was 31.1 (SD = 5.73). About half of the respondents (50.1%) completed university education, followed by those who attended secondary schools (34.3%). Nearly, three-quarters of the respondents (74.7%) were housewives. About 79.0% of the respondents self-ranked their income level as a medium. The majority of the respondents mentioned that they either have one or two children (44% and 45.9%, respectively).

Moreover, we found that measles vaccine uptake was highly associated with the mother’s employment, as self-employed mothers were more likely to only partially or not vaccinate their children, followed by mothers who were workers and housewives (*p*-value < 0.017). The number of children was associated with measles vaccine uptake, as families with three and more children were more likely to only partially or not vaccinate their children with measles vaccine compared to mothers with one child (*p*-value = 0.041), see Table 1. Additionally, the study found that perceived accessibility and availability of measles vaccine was associated with measles vaccine uptake, as those who strongly disagreed, disagreed, or were unsure were more likely to partially or not to vaccinate their children (*p*-value = 0.045), see Table 1.

### 3.2. PACV Survey Analysis

A summary of the PACV survey items is shown in (Appendix A), which is divided into three subscales: immunization behavior (items 1 and 2), perceptions of safety and efficacy (items 7–10), and general attitudes and trust (items 3–6 and 11–15). Regarding immunization behavior, 89 (17.8%) respondents delayed getting their child vaccinated for reasons other than illness or allergies. Based on data about perceived safety and efficacy, 16.6% of respondents (very) expressed concern that the measles vaccine might not be effective (item 10), 13% expressed concern that the vaccine might not be safe (item 9), and 19% expressed concern (very) that the vaccine might have serious side effects. Only 3.6% of participants evaluated themselves as being very hesitant about childhood measles shots in terms of their general attitude and trust. Almost all of the respondents (96.2%) said they trust the information they are given about measles vaccinations, and most (88%) said they trust their doctor.

### 3.3. Mediation Analysis

Mediation analysis was conducted using two models, the ordinary least squares path analysis and multiple logistic regression. It shows that the parental perception of the accessibility of the measles vaccine influences the uptake of the measles vaccine indirectly through the mediation effect of measles vaccine hesitancy. As shown in Figure 2 and Table 2, after controlling for the other factors, including the mother’s age and a number of children, parents who perceive the measles vaccine as difficult to access would be more likely to be hesitant towards the measles vaccine (a = 2.7756, *p* < 0.0001).

Although there was no direct association between the parental perception about accessibility to the measles vaccine and uptake of the measles vaccine (c = 0.1236, *p* = 0.4328), hesitancy toward the measles vaccine was found as a predictor of measles vaccine uptake (b = −0.0530 (*p* < 0.0001). The bootstrap confidence intervals for the mediating effects of measles vaccine hesitancy on measles vaccine uptake based on 10,000 bootstrap samples did not include zero, therefore there is a significant association between them. The mediating effects of vaccine hesitancy were Boot 95% CI = 0.0703, 0.2478.

## 4. Discussion

Parental hesitancy toward the measles vaccine is rapidly increasing across different populations in LMICs, potentially reversing decades of progress toward measles elimination in many countries. Understanding parental attitudes toward the measles vaccine and perceived access barriers to vaccinating their children could help the Federal Ministry of Health and other relevant institutions address these issues ahead of time and boost measles vaccine uptake in Sudan. The present study aimed to evaluate whether the measles vaccine uptake can be predicted directly or indirectly by parental perceptions about accessibility to measles vaccination services with potential mediation of measles vaccine hesitancy (PACV scores). To our knowledge, this study is the first of its own kind in the field in Sudan.

### 4.1. Perceived Availability of the Measles Vaccine—Measles Vaccine Hesitancy (PACV)

Our study showed that parents who perceive that the measles vaccine is not accessible would be more likely to be hesitant towards measles vaccination (*p* < 0.0001). This finding is supported by many studies in Africa, as measles vaccination sessions in many countries are conducted as one or two sessions per week or month to comply with the recommendation of discarding the ten-dose vial after six hours from opening the vial as well as reducing the loss of unused doses (i.e., due to the multi-dose vial policy). This may cause vaccine hesitancy as parents are actively trying to get their child vaccinated with the vaccine, but are turned away when the provider refuses to open the measles vaccine vial [7,8,12]. Analysis for immunization policy as well as cost-effective analysis is much needed to anticipate what will happen if the ten-dose vial of measles vaccine is shifted to a five-dose vial in Sudan. Additionally, developing a parental reminding system may positively impact the timely uptake of the measles vaccine [7,28].

### 4.2. PACV—Uptake of the Measles Vaccine

Our study underscored that parental hesitancy toward the measles vaccine influences the uptake of the measles vaccine among children (*p* < 0.0001). As the present study was conducted in a low-income context, it supports the predictive validity of the PACV survey in determining the vaccination status of children, as shown in previous studies in high- and middle-income countries [25,29,30,31,32]. This result highlights the urgent need for developing context-appropriate strategies to mitigate hesitancy towards measles vaccination and improve vaccine uptake in Sudan. To inform decisions and improve vaccination uptake, a variety of strategies could be used to educate parents, caregivers, and communities on child vaccination. These strategies include interventions in which information is aimed at larger groups of people in the community, such as at public meetings, radio, or leaflets. We recommend these communication strategies be enhanced to address the causes of hesitancy towards measles and other vaccines in Sudan.

### 4.3. Perceived Accessibility to the Measles Vaccine—Uptake of the Measles Vaccine

Contrary to the findings from different African countries [8,11,12], measles vaccine accessibility perception was not associated with measles vaccine uptake after controlling other potential social confounders, such as the mother’s age and the number of children. Vaccine access is determined by a variety of economic and political factors, as well as the quality of healthcare systems and their ability to reach every corner of society. As a result, contextual and socio-cultural differences across countries are critical in explaining different patterns of vaccination uptake.

#### Limitations

We acknowledge some limitations related to our study; therefore, the study’s findings should be interpreted within the context of this study. One of the limitations of this study is that we included only parents of 2–3-year-old children, who may be already vaccine hesitant. In addition, there was unintentional selection bias due to the sampling technique (consecutive sampling) to recruit the participants until the pre-determined sample size of the study is reached from both districts. Moreover, the results of the current study are prone to recall bias, as only less than half of the participants (42.8%) showed their children’s vaccination cards during the data collection, which may underestimate the non-vaccination rate. However, concerning the purpose of the current study, we think that these limitations did not have a substantial impact on the main study findings.

## 5. Conclusions

Our main findings underscored that parental perception about accessibility to measles vaccination services is likely to influence the uptake of measles vaccine through the mediating effects of measles vaccine hesitancy. When someone perceives that the measles vaccination is not available, this will not affect his/her child’s uptake of the measles vaccine unless s/he is also vaccine hesitant. In light of these findings, we suggest that intervening in measles vaccine hesitancy will have a direct impact on the uptake of the measles vaccine in Sudan. Hesitancy itself and its determinants should be addressed with effective strategies. Analysis for immunization policy as well as cost-effective analysis is much needed to anticipate what will happen if the ten-dose vial of measles vaccine is shifted to a five-dose vial in Sudan.

## Figures and Tables

**Figure 1 vaccines-10-01674-f001:**
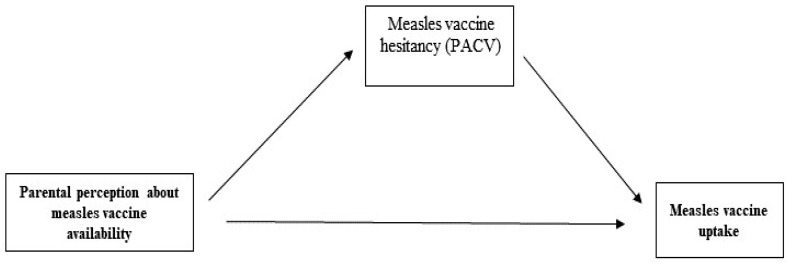
Mediation with the path of Measles Vaccine Hesitancy (PACV scores) as the mediator in the relationship between parental perception about measles vaccine availability and measles vaccine uptake.

**Figure 2 vaccines-10-01674-f002:**
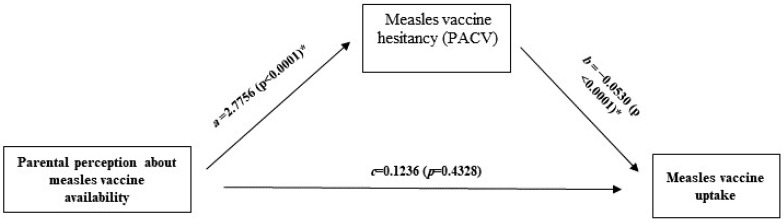
Mediation with path (β) coefficients and (*p*-value) of Measles Vaccine Hesitancy (PACV) as the mediator in the relationship between parental perception about measles vaccine availability and measles vaccine uptake. * Statistically significant.

**Table 1 vaccines-10-01674-t001:** Association of the social and behavioral characteristics of the surveyed parents/guardians with measles vaccination status. N = 495.

Variables		Measles Vaccination Uptake/Status					
		Total N = 495 (%)	Fully Vaccinated		Partially/Unvaccinated		*p*-Value
			N = 436	%	N = 59	%	
Area of the study	Alsharafia	152 (30.7)	128	84.2%	24	15.8%	0.077
	Abo Saeed	343 (69.3)	308	89.8%	35	10.2%	
Mother’s Education	Illiterate	14 (2.8)	11	78.6%	3	21.4%	0.162
	Primary completed	63 (12.7)	51	81.0%	12	19.0%	
	Secondary completed	170 (34.3)	151	88.8%	19	11.2%	
	University	248 (50.1)	223	89.9%	25	10.1%	
Mother’s Employment	Housewife	370 (74.7)	323	87.3%	47	12.7%	0.017 *^,b^
	Student	11 (2.2)	10	90.9%	1	9.1%	
	Domestic worker	14 (2.8)	12	85.7%	2	14.3%	
	Officer	50 (10.1)	48	96.0%	2	4.0%	
	Professional (e.g., Engineer, Doctor)	33 (6.7)	32	97.0%	1	3.0%	
	Self-employed	16 (3.2)	10	62.5%	6	37.5%	
	Others	1 (0.2)	1	100.0%	0	0.0%	
Income Level (Self-Ranking)	High	70 (14.1)	65	92.9%	5	7.1%	0.268
	Medium	391 (79.0)	343	87.7%	48	12.3%	
	Low	34 (6.9)	28	82.4%	6	17.6%	
Number of Children	1	218 (44.0)	185	84.9%	33	15.1%	0.041 *
	2	227 (45.9)	209	92.1%	18	7.9%	
	3 and more	50 (10.1)	42	84.0%	8	16.0%	
Perceived accessibility and availability of measles vaccine when my child needs it.	Strongly agree	247 (49.9)	217	87.9%	30	12.1%	0.045 *^,b^
	Agree	201 (40.6)	183	91.0%	18	9.0%	
	Not sure	10 (2.0)	8	80.0%	2	20.0%	
	Disagree	35 (7.1)	27	77.1%	8	22.9%	
	Strongly disagree	2 (0.4)	1	50.0%	1	50.0%	

* Statistically significant. ^b^ Fisher’s exact test.

**Table 2 vaccines-10-01674-t002:** Regression coefficients, standard errors, and model summary information for the parental perception about measles vaccine availability using the mediation model depicted in Figure 2.

	M (Vaccine Hesitancy)			Y (Measles Vaccine Uptake) **		
Antecedent	*Coeff.*	*SE*	*p*	*Coeff.*	*SE*	*p*
X (Perceived vaccine availability)	*a*, 2.7756	0.5411	<0.0001 *	*c*, 0.1236	0.1576	0.4328
M	-	-	-	*b*, 0.0530	0.0124	<0.0001 *
Cov1 (Mothers’ age)	0.0822	0.0802	0.306	0.0432	0.0246	0.0787
Cov2 (N. of children)	1.9258	0.6511	0.003 *	−0.2697	0.2040	0.1862
Constant	−0.9268	2.9151	0.751	−3.7734	0.9146	<0.0001 *
Model Summary	*R-sq* = 0.0718, *F* = 12.551, *p* < 0.0001 *					

* Statistically significant. ** Multiple Logistic Regression Model.

## Data Availability

The data presented in this study are available from the corresponding author upon request.

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
