# Peer review of "Perceived Vaccine Availability and the Uptake of Measles Vaccine in Sudan: The Mediating Role of Vaccination Hesitancy"

_vaccines, 2022, doi:10.3390/vaccines10101674_

Round 1

Reviewer 1 Report

The article written by Majdi M and co-authors presents a study on vaccination hesitancy for Measles vaccines in Sudan by cross-sectional design.

Although, it is an interesting work, there are number of shortcomings.

Abstract needs to be formatted as per journal template.

The inclusion of word ‘uptake’ is not clear. Actually uptake refers to “an act or instance of absorbing or incorporating something especially into a living organism, tissue, or cell” like uptake of glucose/substances into cells..

If it refers to acceptability in this study, there should be alternate term to uptake.

The details about the primary outcome and independent variables needs to be mentioned in brief in abstract section.

It is also not clear whether study focused on access issues or vaccine acceptability by parents? Unfortunately, I could not find any item to evaluate access issues in the abstract section. If it is just speculative, it can’t be included in the title as it is done now.

Figure 1 is not clear, the boxes states differently from the explanation given in the description of figure. PACV is given in box as hesitancy, while it is used as attitude in description. Also, attitude and perception are used synonymously in this description and box statement. It should be consistent; it is not same.

What is the use of lines in introduction from 40 to 53 about dosing and vials? How it is relevant? Even if it is, is it the only factors that needs to be discussed so much. Summarization is enough, maybe add other factors on accessibility issues.

The study is old done in February 2019, how relevant it is today, lot of changes have happened especially with relevant to vaccines and its hesitancies.

As per the section 2.3.2, some parents produced card, some were not. Was their any comparison done among them on hesitancy or intake of vaccine.

Why authors inquired only about mother’s education and employment? Why not father? Especially, when your significant acceptability among those mothers who are employed. Maybe, father’s employment would have given some more direction to the study. Also, how reliable this finding when three quarter of the mother were not working.

What about the availability of vaccine in Sudan? Is it free or parents have to pay? That may be important factor for acceptability.

How did authors used Parents Attitude about Childhood Vaccination in Sudan? Was this used in English or translated. More details are required about translation if it was done.

Also, need details about method of administering this questionnaire.

Include IRB approval number with date.

Many confounding factors missed in the study, that include the vaccine supply system, status of other vaccination, the role of community, family, media, political system, government regulations, mother’s health status and father’s education, and involvement. It is not possible to develop credibility on the study based on only perception from mother. Although, they play important role, father’s decision is equally necessary. Also, why authors selected daytime it is a working time for fathers? What is the credibility of this data collectors? How much they knew about the study?

The following conclusion statement needs re-writing…..If someone perceived that measles vaccination is not accessible, this will not influence his child’s uptake of the measles vaccine 297 unless s/he is hesitant too towards the measles vaccine.

Author Response

Response to Reviewer 1 Comments

Abstract needs to be formatted as per journal template.

Response: The abstract is now formatted using the journal template.

The inclusion of word ‘uptake’ is not clear. Actually uptake refers to “an act or instance of absorbing or incorporating something especially into a living organism, tissue, or cell” like uptake of glucose/substances into cells..

If it refers to acceptability in this study, there should be alternate term to uptake.

Response: We would argue that ‘Vaccine uptake’ as a term is widely used in WHO documents, here are some relevant references:

  • World Health Organization. Report of the SAGE working group on vaccine hesitancy. Geneva, Switzerland: WHO. 2014 Nov.
  • Behavioural and social drivers of vaccination: tools and practical guidance for achieving high uptake. Geneva: World Health Organization; 2022. Licence: CC BY-NC-SA 3.0 IGO.

The details about the primary outcome and independent variables needs to be mentioned in brief in abstract section.

Response: The primary outcome and independent variables were briefly described in the abstract.

It is also not clear whether study focused on access issues or vaccine acceptability by parents? Unfortunately, I could not find any item to evaluate access issues in the abstract section. If it is just speculative, it can’t be included in the title as it is done now.

Response: This study focused on the perceived access issues and its impact on vaccine impact. Physical availability of vaccine is considered as one of the ‘’Access issues’’ as described in SAGE Report in 2014 [1], and as part of the ‘’Practical issue’’ in the BeSD framework that adopted by WHO in May 2022 [2]. In this study we evaluated the perceived availability among parents rather than (the real) availability. To be more specific, we changed the title from general access issue to perceived vaccine availability.

References:

[1] World Health Organization. Report of the SAGE working group on vaccine hesitancy. Geneva, Switzerland: WHO. 2014 Nov.

[2] WHO. Behavioural and social drivers of vaccination: tools and practical guidance for achieving high uptake. Geneva: World Health Organization; 2022. Licence: CC BY-NC-SA 3.0 IGO.

Figure 1 is not clear, the boxes states differently from the explanation given in the description of figure. PACV is given in box as hesitancy, while it is used as attitude in description. Also, attitude and perception are used synonymously in this description and box statement. It should be consistent; it is not same.

Response: We revised and corrected the description of Figure 1 as you suggested.

What is the use of lines in introduction from 40 to 53 about dosing and vials? How it is relevant? Even if it is, is it the only factors that needs to be discussed so much. Summarization is enough, maybe add other factors on accessibility issues.

Response: Thank you for giving us the concrete direction. We revised  summarized the this point as you suggested.

The study is old done in February 2019, how relevant it is today, lot of changes have happened especially with relevant to vaccines and its hesitancies.

Response: Given the study’s findings in early 2019, which showed the complexity of measles vaccine availability, hesitancy and uptake in Sudan, we expect that the current evolving COVID-19 pandemic worldwide, and particularly in Sudan, where, the vaccination services that provide routine vaccines including the measles vaccine have been disrupted for a while, along with reports of hesitancy towards COVID-19 vaccines in Sudan, will worsen the rate of uptake of measles vaccines in Sudan.

As per the section 2.3.2, some parents produced card, some were not. Was their any comparison done among them on hesitancy or intake of vaccine.

Response: Thanks for raising this point. We did not conduct further analysis to compare between those who showed their cards and those who did not. But we found that only less than half of the participants (42%) showed their cards and we added this point as in the limitation section,

Why authors inquired only about mother’s education and employment? Why not father? Especially, when your significant acceptability among those mothers who are employed. Maybe, father’s employment would have given some more direction to the study. Also, how reliable this finding when three quarter of the mother were not working.

Response: Thanks for raising this point.  As we reported in the limitation that from a gender perspective, we missed fathers’ perspective in this study, as data was collected during the work time. Additionally, we wanted to investigate the link between the individual perception about the availability with the measles uptake by their children. We found the majority of the participants were mothers (87.2%)  as described in another relevant article [1]. Moreover, Sudanese mothers, as in many other African countries, are mostly the first persons responsible for the health and prevention of sickness of their children/family and should know the health situation best. [2]

References:

[1] Sabahelzain MM, Dubé E, Moukhyer M, Larson HJ, van den Borne B, Bosma H. Psychometric properties of the adapted measles vaccine hesitancy scale in Sudan. PLoS One. 2020 Aug 6;15(8):e0237171. doi: 10.1371/journal.pone.0237171. PMID: 32760162; PMCID: PMC7410231.

[2] Caldwell J.C., Caldwell P. Roles of Women, Families, and Communities in Preventing Illness and Providing Health Services in Developing Countries. In: National Research Council (US) Committee on PopulationGribble J.N., Preston S.H., editors. The Epidemiological Transition: Policy and Planning Implications for Developing Countries: Workshop Proceedings. National Academies Press (US); Washington, DC, USA: 1993.

What about the availability of vaccine in Sudan? Is it free or parents have to pay? That may be important factor for acceptability.

Response: Thanks for this point. Vaccines are provided free of charge in the primary healthcare centers (PHC). But, for measles vaccination only 1-2 sessions are allowed per week which lead to perceiving unavailability of the vaccine.

How did authors used Parents Attitude about Childhood Vaccination in Sudan? Was this used in English or translated. More details are required about translation if it was done.

Response: We added a description about the reliability and validity of PACV in Sudan: (Lines 151-154): ‘’The Arabic version of the PACV scale was pre-tested with 16 parents at the Ahfad Family Center (Primary health care center) to assess its understandability and clarity of questions. The reliability and validation test was carried out and described by Sabahelzain et al. The Cronbach's alpha was computed for this scale (Q3-Q15) which is 0.62’’

Also, need details about method of administering this questionnaire.

Response: We described briefly the method of administering the questionnaire in Lines (124-125) ‘’The questionnaire was in Arabic language and self-administered, but when needed, the data collectors helped the participants to fill out the questionnaire.’’

Include IRB approval number with date.

Response: The IRB approval number with date were included.

Many confounding factors missed in the study, that include the vaccine supply system, status of other vaccination, the role of community, family, media, political system, government regulations, mother’s health status and father’s education, and involvement. It is not possible to develop credibility on the study based on only perception from mother. Although, they play important role, father’s decision is equally necessary. Also, why authors selected daytime it is a working time for fathers? What is the credibility of this data collectors? How much they knew about the study?

Response: We agree with the reviewer that there are many factors influencing vaccine uptake, some of which are related to supply side and the other related to demand side which were addressed in many studies and reports. However, perception of vaccine availability was not well studied especially in Sudan, as well as the role of vaccine hesitancy as mediating factor. We already acknowledged many challenges as limitations in the limitation section, including missing out the fathers’ perspectives. The data collectors were final year students of school of health sciences. The field data collection was supervised by the first author as part of his PhD research project in Maastricht University.

The following conclusion statement needs re-writing…..If someone perceived that measles vaccination is not available or accessible, this will not influence his child’s uptake of the measles vaccine unless s/he is hesitant too towards the measles vaccine.

Response: We revised and re-write/paraphrased the statement as you suggested.  ‘When someone perceives that measles vaccination is not available, this will not affect his child's uptake of the measles vaccine unless s/he is hesitant too.’’

Reviewer 2 Report

Intro

1. There seems to be a contradiction on lines 35 to 39. If studies have found how access impacts acceptance and demand, then doesnt that also impact uptake. Please clarify this

2. Lines 40-45 is a run on sentence. Please fix

3. I think you are taking the SAGE definition of vaccine hesitancy too literally. The focus should be on people that refuse vaccine despite its availability, but you seem to be saying that vaccine hesitancy also occurs when providers dont open a vial despite people present wanting vaccine. Vaccine hesitancy is on the patient/caregiver, not the provider. There are several mentions of this throughout your paper that need to be clarified and corrected. Vaccine hesitancy doesnt seem applicable in your study because it seems people are present at clinics waiting to get vaccinated since that is where you are surveying them. 

4. Line 59. SAGE should be in parenthesis, e.g. (SAGE) 

5. Lines 64-72: I think this paragraph is what you should use to open up your study. Illustrate the need to improve measles vaccinations rates and the reasons why Further, how does the vaccination rate in Sudan compare to other African countries. 88% for first dose and 72% for second dose doesnt seem that bad all things considering (despite vaccination needing to be at 90% to avoid outbreaks). 

6. Lines 67-70: arent these previous studies that you cite also the focus of your study? Is your study a me-too study? Please clarify. This seems to be another contradiction in your paper. 

7. Figure 1 is not mentioned in the text nor has any description of the figure, why its included, relavence, etc. 

Methods

1. You are including parents of children 2-3 years of age in your study, yet in Sudan, the vaccination series for measles starts at 9 months of age. Isnt this a major limitation of your study as parents of 2-3 year olds would already be vaccine hesitant to one degree or another to begin with? This needs to be called out in your limitations section. 

2. Line 104: references 10 and 30 are for studies in Tennessee and Tanzania, not Sudan. I believe your citation numbers are all incorrect and need to be fixed and updated. I would be interested to better understand this larger study that your paper references  and how this current study fits into it. 

3. Line 133: If you asked parents about their childs measles vaccination status if they didnt have a vaccine card, isnt that also a limitation of your study as their may be recall bias? Please clarify/correct. 

4. Lines 139-141: several times throughout the paper, you reference this sentence, but after reading it many times, I really do not understand what it means. Please clarify throughout your paper. 

5. It seems that you used the PACV survey to measure vaccine hesitancy, but according to lines 148 and 149, it sounds like you used the results in a different way than what the survey intended to be used for? If that is the case, you should also explain how your results are valid (if deviating from how you would normally total up a score on the PACV). 

6. Line 154: mention of Cronbach's alpha is a result that belongs in the result section, not in methods. Please fix. 

Results

1. Half of page five, beyond line 205 is missing. I am not sure if there is/was a file upload issue, but it appears that text is missing from your paper. 

2. There is no mention anywhere of results from the PACV survey, expect in the appendix. Many times in your paper,  you mention patient perception, but you have not really presented any data that discussed this in any great detail. An analysis of the PACV survey needs to be included and featured in your paper if this paper is to be published. 

3. Lines 214-217 and lines 218-221. This appears to be another contradiction in your paper. Please review and clarify these sentences. As is, they conflict with one another. 

Discussion

1. The statement that patients who perceive the measles vaccine is not accessible would be more likely to be hesitant towards measles vaccination has no basis in the data that you presented and again conflicts with SAGE's  defination of vaccine hesitancy that you have included. 

2. Lines 251-254 ("analysis for immunization policy...") and lines 264-276 ("these strategies include.."): these should be part of your conclusions and not discussion. Discussion section is for you to explain the results you got in your study. 

Limitations

1. Again, the main limitation of this work is that you have included parents of 2-3 year olds in the study. This patient population is likely already vaccine hesitant and needs to be included.

2. How is selection bias present in your study if you conducted a conveinence sampling of patients? Please clarify 

3. Lines 281-283 ("additionally, our study did not show..."): this is NOT a limitation. It is just a result of your work and needs to be removed here

4. Recall bias should also be included as a limitation as mentioned above. 

Conclusions

1. First paragraph of is yet another contradiction. Please fix. 

Author Response

Response to Reviewer 2 Comments

Introduction

  1. There seems to be a contradiction on lines 35 to 39. If studies have found how access impacts acceptance and demand, then doesn’t that also impact uptake. Please clarify this

Response: Thank you very much for this comment. These sentences were revised for more clarity. Amendments were highlighted with an active track-changes function.

  1. Lines 40-45 is a run on sentence. Please fix

Response: We revised according to the reviewer comment.

  1. I think you are taking the SAGE definition of vaccine hesitancy too literally. The focus should be on people that refuse vaccine despite its availability, but you seem to be saying that vaccine hesitancy also occurs when providers dont open a vial despite people present wanting vaccine. Vaccine hesitancy is on the patient/caregiver, not the provider. There are several mentions of this throughout your paper that need to be clarified and corrected. Vaccine hesitancy doesn’t seem applicable in your study because it seems people are present at clinics waiting to get vaccinated since that is where you are surveying them. 

Response: Thank you for this comment. As the current study investigated the impact of perceived availability (not real), this was mentioned as an example of access issues. In addition, this study was a community-based survey not included parents attending child vaccination sessions. These sentences were revised and clarified.

  1. Line 59. SAGE should be in parenthesis, e.g. (SAGE) 

Response: Corrected.

  1. Lines 64-72: I think this paragraph is what you should use to open up your study. Illustrate the need to improve measles vaccination rates and the reasons why Further, how does the vaccination rate in Sudan compare to other African countries. 88% for the first dose and 72% for the second dose doesn’t seem that bad all things considering (despite vaccination needing to be at 90% to avoid outbreaks). 

Response: Thank you for giving us the concrete direction. This section was revised according to the reviewer’s comment, we compared the reported vaccination rates in contrast to the 95% level required for measles elimination, entailing that much still needs to be done. Additionally, the coverage was calculated using “Administrative Method” which has many limitations especially in a country high number of internally displaced people and migration.

Lines 67-70: arent these previous studies that you cite also the focus of your study? Is your study a me-too study? Please clarify. This seems to be another contradiction in your paper. 

Response: This was revised as mentioned in the first comment.

  1. Figure 1 is not mentioned in the text nor has any description of the figure, why its included, relavence, etc. 

Response: This was revised and corrected

Methods

  1. You are including parents of children 2-3 years of age in your study, yet in Sudan, the vaccination series for measles starts at 9 months of age. Isnt this a major limitation of your study as parents of 2-3 year olds would already be vaccine hesitant to one degree or another to begin with? This needs to be called out in your limitations section. 

Response: We would argue that this is not a major limitation because the eligibility criteria for parents is to have a child aged (2-3) years to guarantee that the children were eligible to have their two doses of measles at age 9 and 18 months when we assess their current vaccination status which is the main outcome of this study. However, because of uncertainty, we added this valued suggestion. 

Response: The limitation section was revised and all possible limitations were mentioned.

  1. Line 104: references 10 and 30 are for studies in Tennessee and Tanzania, not Sudan. I believe your citation numbers are all incorrect and need to be fixed and updated. I would be interested to better understand this larger study that your paper references and how this current study fits into it. 

Response: Thank you for this important notice. We revised the references and correct references were inserted.

  1. Line 133: If you asked parents about their childs measles vaccination status if they didnt have a vaccine card, isnt that also a limitation of your study as their may be recall bias? Please clarify/correct. 

Response: The limitation section was revised and all possible limitations were mentioned including this point as you raised.

  1. Lines 139-141: several times throughout the paper, you reference this sentence, but after reading it many times, I really do not understand what it means. Please clarify throughout your paper. 

Response: This section about the measurement of the dependent variable was revised according to the reviewer’s comment. Changes made were highlighted  within the revised version.  

  1. It seems that you used the PACV survey to measure vaccine hesitancy, but according to lines 148 and 149, it sounds like you used the results in a different way than what the survey intended to be used for? If that is the case, you should also explain how your results are valid (if deviating from how you would normally total up a score on the PACV). 

Response: Thanks for this observation, we used the PACV survey for the same intended purpose. Therefore, we revised this part (line 148-149) to avoid any confusion.

  1. Line 154: mention of Cronbach's alpha is a result that belongs in the result section, not in methods. Please fix. 

Response: Thank you for this comment, however, we prefer mentioning the results of the reliability test in this section for more clarity to the readers, as presenting Cronbach Alpha within the results would give a hint that this was one of the objectives of the study.

Results

  1. Half of page five, beyond line 205 is missing. I am not sure if there is/was a file upload issue, but it appears that text is missing from your paper. 

Response: This was due to formatting Table 1. There were no missing sentences in the uploaded version of the manuscript. 

There is no mention anywhere of results from the PACV survey, expect in the appendix. Many times in your paper,  you mention patient perception, but you have not really presented any data that discussed this in any great detail. An analysis of the PACV survey needs to be included and featured in your paper if this paper is to be published.

Response: Now, we added a subsection about PACV survey’s analysis in the result section 3.2. PACV survey analysis: A summary of the PACV survey items is shown in (Appendix A), which is divided into three subscales: immunization behavior (items 1 and 2), perceptions of safety and efficacy (items 7–10), and general attitudes and trust (items 3–6 and 11–15). Regarding immunization behavior, 89 (17.8%) respondents delay getting their child vaccinated for reasons other than illness or allergies. Based on data about perceived safety and efficacy, 16.6% of respondents (very) expressed concern that the measles vaccine might not be effective (item 10), 13% expressed concern that the vaccine might not be safe (item 9), and 19% expressed concern (very) that the vaccine might have serious side effects. Only 3.6% of participants evaluated themselves as being very hesitant about childhood measles shots in terms of their general attitude and trust. Almost all of the respondents (96.2%) said they trust the information they are given about measles vaccinations, and most (88%) said they trust their doctor.”

  1. Lines 214-217 and lines 218-221. This appears to be another contradiction in your paper. Please review and clarify these sentences. As is, they conflict with one another. 

Response: We clarified this section according to the reviewer’s comment. In this sentence, we reported that no (direct) association was found between perceived access issues and the uptake of the vaccine, however, there was a significant association between the uptake of the vaccine and measles vaccine hesitancy.

Discussion

The statement that patients who perceive the measles vaccine is not accessible would be more likely to be hesitant towards measles vaccination has no basis in the data that you presented and again conflicts with SAGE's  defination of vaccine hesitancy that you have included. 

Response: We would argue that our data in table 2 showed perceived vaccine accessibility and availability is strongly associated with vaccine hesitancy (p. value< 0.0001). Additionally, in the vaccine hesitancy definition by SAGE, “Convenience” is considered as one of the 3Cs Model/ factors that influence vaccine hesitancy. Convenience was defined by the same group as Extent to which physical availability, affordability, willingness-to-pay for, geographical accessibility, ability to understand (language and health literacy) and appeal of immunization services affects uptake.”  So, even our finding is supported by the definition of SAGE.

References:

  • World Health Organization. Report of the SAGE working group on vaccine hesitancy. Geneva, Switzerland: WHO. 2014 Nov.
  • Bedford H, Attwell K, Danchin M, Marshall H, Corben P, Leask J. Vaccine hesitancy, refusal and access barriers: The need for clarity in terminology. Vaccine. 2018 Oct 22;36(44):6556-6558. doi: 10.1016/j.vaccine.2017.08.004. Epub 2017 Aug 19. PMID: 28830694

  1. Lines 251-254 ("analysis for immunization policy...") and lines 264-276 ("these strategies include.."): these should be part of your conclusions and not discussion. Discussion section is for you to explain the results you got in your study. 

Response: This was revised according to the reviewer’s comment. we mentioned the need for policy analysis and the development of successful strategies to fight vaccine hesitancy.

Limitations

  1. Again, the main limitation of this work is that you have included parents of 2-3 year olds in the study. This patient population is likely already vaccine hesitant and needs to be included.

Response: This was included in the limitation section.

  1. How is selection bias present in your study if you conducted a convenience sampling of patients? Please clarify 

Response: As the sample selection was not based on random selection, this entails selection bias.

  1. Lines 281-283 ("additionally, our study did not show..."): this is NOT a limitation. It is just a result of your work and needs to be removed here

Response: Removed.

  1. Recall bias should also be included as a limitation as mentioned above. 

Response: Revised and included.

Conclusions

  1. First paragraph of is yet another contradiction. Please fix. 

Response: The conclusion section was revised accordingly.

Reviewer 3 Report

Thank you for the invitation to review this manuscript. This manuscript carries important implications, however; I have a few concerns here.

The PVAC scale was used to evaluate the primary outcome of this study. I am not sure whether the scale was translated or not. Although PVAC is a validated tool, its translation into the Arabic language will further require the validation and reliability process. Moreover, how the translation was done is not clear from the study. It is suggested to provide this information clearly and upload the English and Arabic data collection form as a supplementary file.

Please provide a brief description of mediation analysis, as I have noticed that most of the young researchers and students are still not much familiar with this type of analysis. It enhances the educational impact of this manuscript. 

The authors should provide a summary of the articles, previously published on this topic in Sudan, if no; the authors should mention that this study is first of its own kind in the field.

Author Response

Response to Reviewer 3 Comments

The PVAC scale was used to evaluate the primary outcome of this study. I am not sure whether the scale was translated or not. Although PVAC is a validated tool, its translation into the Arabic language will further require the validation and reliability process. Moreover, how the translation was done is not clear from the study. It is suggested to provide this information clearly and upload the English and Arabic data collection form as a supplementary file.

 Response: We added a description about the reliability and validity of PACV in Sudan: (Lines 151-154): ‘’The Arabic version of the PACV scale was pre-tested with 16 parents at the Ahfad Family Center (Primary health care center) to assess its understandability and clarity of questions. The reliability and validation test was carried out and described by Sabahelzain et al. The Cronbach's alpha was computed for this scale (Q3-Q15) which is 0.62’’

Please provide a brief description of mediation analysis, as I have noticed that most of the young researchers and students are still not much familiar with this type of analysis. It enhances the educational impact of this manuscript. 

Response: a brief description of simple mediation analysis was added in lines 170-172: ‘’ Hayes describes the objective of doing simple mediation model as ‘to demonstrate how a variable’s effect on an outcome can be partitioned into direct and indirect effects that can be quantified..’’

The authors should provide a summary of the articles, previously published on this topic in Sudan, if no; the authors should mention that this study is first of its own kind in the field.

 Response: There are some relevant studies about measles vaccine hesitancy were conducted in Sudan and were already described in the manuscript, but we added this statement ‘To our knowledge, this study is the first of its own kind in the field in Sudan’’

Round 2

Reviewer 1 Report

This manuscript has undergone substantial changes. I am convinced with the revision. They addressed my concerns. I recommend it for acceptance now.